# Association between non-malignant monoclonal gammopathy and adverse outcomes in chronic kidney disease: A cohort study

Anthony Fenton[1,2]*, Rajkumar Chinnadurai[3], Latha Gullapudi[4], Petros Kampanis[5], Indranil Dasgupta[1], James Ritchie[3], Stephen Harding[5], Charles J. Ferro[1], Philip A. Kalra[3], Maarten W. Taal[4], Paul Cockwell[1,2]

1 University Hospitals Birmingham NHS Foundation Trust, Birmingham, United Kingdom, 2 Institute of Inflammation and Ageing, University of Birmingham, Birmingham, United Kingdom, 3 Salford Royal NHS Foundation Trust, Salford, United Kingdom, 4 Division of Medical Sciences and Graduate Entry Medicine, School of Medicine, University of Nottingham, Nottingham, United Kingdom, 5 Binding Site, Birmingham, United Kingdom

* anthony.fenton1@nhs.net

**Data Availability Statement:** Due to ethical restrictions by the National Research Ethics Service (NRES) Committee West Midlands-South

## Abstract

### Background

In studies including the general population, the presence of non-malignant monoclonal gammopathy (MG) can be causally associated with kidney damage and shorter survival. We assessed whether the presence of an MG is associated with a higher risk of kidney failure or death in individuals with chronic kidney disease (CKD).

### Methods and findings

Data were used from 3 prospective cohorts of individuals with CKD (not on dialysis or with a kidney transplant): (1) Renal Impairment in Secondary Care (RIISC, Queen Elizabeth Hospital and Heartlands Hospital, Birmingham, UK, $N = 878$), (2) Salford Kidney Study (SKS, Salford Royal Hospital, Salford, UK, $N = 861$), and (3) Renal Risk in Derby (RRID, Derby, UK, $N = 1,739$). Participants were excluded if they had multiple myeloma or any other B cell lymphoproliferative disorder with end-organ damage. Median age was 71.0 years, 50.6% were male, median estimated glomerular filtration rate was 42.3 ml/min/1.73 m$^2$, and median urine albumin-to-creatinine ratio was 3.4 mg/mmol. All non-malignant MG was identified in the baseline serum of participants of RIISC. Further, light chain MG (LC-MG) was identified and studied in participants of RIISC, SKS, and RRID. Participants were followed up for kidney failure (defined as the initiation of dialysis or kidney transplantation) and death. Associations with the risk of kidney failure were estimated by competing-risks regression (handling death as a competing risk), and associations with death were estimated by Cox proportional hazards regression. In total, 102 (11.6%) of the 878 RIISC participants had an MG. During a median follow-up time of 74.0 months, there were 327 kidney failure events and 202 deaths. The presence of MG was not associated with risk of kidney failure

Birmingham, data cannot be shared publicly and is subject to adherence to existing ethics approval. Interested, qualified researchers can request the data by contacting Penelope Gregory, REC Manager, at NRESCommittee.westmidlands-southbirmingham@nhs.net.

**Funding:** The authors received no specific funding for this work.

**Competing interests:** I have read the journal's policy and the authors of this manuscript have the following competing interests: PC has consulted for and advised the Binding Site, who produce the Freelite assay that has been used to measure the LC MGUS reported in this paper, and who carried out the intact immunoglobulin assays. SH is on the Board of Directors for The Binding Site who produce the Freelite assay used in this paper. MWT is a member of the Editorial Board of *PLOS Medicine*.

**Abbreviations:** ACR, albumin-to-creatinine ratio; CKD, chronic kidney disease; COPD, chronic obstructive pulmonary disease; DM, diabetes mellitus; eGFR, estimated glomerular filtration rate; FLC, free light chain; HR, hazard ratio; IHD, ischaemic heart disease; KRT, kidney replacement therapy; LC-MG, light chain monoclonal gammopathy; MAP, mean arterial pressure; MG, monoclonal gammopathy; MGRS, monoclonal gammopathy of renal significance; MGUS, monoclonal gammopathy of undetermined significance; PAD, peripheral artery disease; RIISC, Renal Impairment in Secondary Care; RRID, Renal Risk in Derby; SHR, subhazard ratio; SKS, Salford Kidney Study; SPEP, serum protein electrophoresis.

(univariable subhazard ratio [SHR] 0.97 [95% CI 0.68 to 1.38], $P = 0.85$; multivariable SHR 1.16 [95% CI 0.80 to 1.69], $P = 0.43$), and although there was a higher risk of death in univariable analysis (hazard ratio [HR] 2.13 [95% CI 1.49 to 3.02], $P < 0.001$), this was not significant in multivariable analysis (HR 1.37 [95% CI 0.93 to 2.00], $P = 0.11$). Fifty-five (1.6%) of the 3,478 participants from all 3 studies had LC-MG. During a median follow-up time of 62.5 months, 564 of the 3,478 participants progressed to kidney failure, and 803 died. LC-MG was not associated with risk of kidney failure (univariable SHR 1.07 [95% CI 0.58 to 1.96], $P = 0.82$; multivariable SHR 1.42 [95% CI 0.78 to 2.57], $P = 0.26$). There was a higher risk of death in those with LC-MG in the univariable model (HR 2.51 [95% CI 1.59 to 3.96], $P < 0.001$), but not in the multivariable model (HR 1.49 [95% CI 0.93 to 2.39], $P = 0.10$). An important limitation of this work was that only LC-MG, rather than any MG, could be identified in participants from SKS and RRID.

## Conclusions

The prevalence of MG was higher in this CKD cohort than that reported in the general population. However, the presence of an MG was not independently associated with a significantly higher risk of kidney failure or, unlike in the general population, risk of death.

## Author summary

### Why was this study done?

- Non-malignant monoclonal gammopathy (MG) is a common pre-malignant condition that can be associated with kidney damage and a higher risk of death.

- Chronic kidney disease (CKD) is also common, affecting approximately 10% of adults, and people with CKD may be more likely to have an MG.

- It is unknown whether having an MG is associated with higher risk of kidney failure or death in people with CKD.

### What did the researchers do and find?

- We examined how common an MG is in patients with CKD, and also examined the associations between MG and kidney failure and death using data from 3 cohort studies of patients with CKD in the UK.

- An MG was present in 11.6% of patients with CKD.

- Patients with an MG did not have a higher risk of kidney failure or death.

### What do these findings mean?

- Our results are consistent with a previous study that suggested that MG is more common in patients with CKD.

- However, the presence of an MG was not associated with a higher risk of kidney failure or death.

- Although this is reassuring for patients with CKD and their healthcare providers, we did not examine whether having an MG is associated with the risk of other adverse outcomes.

## Introduction

Monoclonal gammopathy (MG) is a prevalent condition [1] that can indicate a malignant B cell disorder such as multiple myeloma, but more often results from a pre-malignant B cell clone with no demonstrable organ damage (MG of undetermined significance, MGUS) [2,3]. MGUS is common, affecting around 3% of those aged over 50 years [1], and requires monitoring but not targeted therapy. However, a non-malignant MG can, rarely, be causally associated with kidney damage (MG of renal significance [MGRS]), where specific targeted therapy is indicated to preserve organ function [4].

Several studies in general populations have shown that the presence of MGUS is associated with shorter survival [5,6], and the prevalence of MGUS in patients with chronic kidney disease (CKD) has been reported to be higher than in the general population [7,8]. However, to the best of our knowledge, there has been only 1 small study examining the prognostic significance of MGUS in patients with CKD [7]. The study found that those with MGUS were not at a significantly higher risk of kidney failure or, unlike in the general population, death [7].

This study was performed to determine the prevalence of non-malignant MG in a CKD population and to evaluate whether the presence of a non-malignant MG adds to the risk of adverse outcomes in CKD. We tested the pre-specified hypotheses that the presence of a non-malignant MG is independently associated with a higher risk of kidney failure and shorter survival in patients with CKD.

## Methods

A prospective analysis plan was written (S1 Protocol). This study is reported as per the Strengthening the Reporting of Observational Studies in Epidemiology (STROBE) guideline (S1 STROBE Checklist).

### Patients

Patients from 3 prospective UK cohorts of individuals with CKD who had not received kidney replacement therapy (KRT, i.e., dialysis or kidney transplant) were included: Renal Impairment in Secondary Care (RIISC; ClinicalTrials.gov: NCT01722383), the Salford Kidney Study (SKS, previously termed the Chronic Renal Insufficiency Standards Implementation Study [CRISIS]), and Renal Risk in Derby (RRID; National Institute for Health Research study ID: 6632).

Each study had research ethics committee (REC) approval (RIISC: West Midlands South Birmingham REC, ref 10/H1207/6; SKS: North West Greater Manchester South REC; ref 15/NW/0818; RRID: East Midlands Nottingham 1 REC). All participants in all 3 studies provided written informed consent, and all studies were conducted in accordance with the Declaration of Helsinki.

Complete details of these cohorts have been reported elsewhere [9–11], and the inclusion and exclusion criteria are summarised in Table 1. For this analysis, participants were excluded

at an individual level if they had a malignant MG (multiple myeloma or another malignant B cell lymphoproliferative disorder).

## Definition of MG

Although the majority of included participants with MG will have had MGUS, we have used the more general term non-malignant MG to reflect the fact that only a minority of participants had kidney biopsies to exclude MGRS definitively. Two forms of non-malignant MG were assessed: (1) any non-malignant MG (assessed in the RIISC cohort only), defined as (a) a monoclonal protein on serum protein electrophoresis (SPEP) confirmed by serum immunofixation or (b) a serum κ/λ free light chain (FLC) ratio < 0.37 or >3.10 with an increased level of the involved light chain, or (2) light chain MG (LC-MG) (assessed in all 3 cohorts), defined as a serum κ/λ FLC ratio < 0.37 or >3.10 with an increased level of the involved light chain.

The Freelite assay (Binding Site, Birmingham, UK) was used to measure κ and λ FLC concentration in all 3 cohorts. The serum κ/λ FLC ratio reference range of 0.37 to 3.10 has been recommended in patients with kidney impairment to account for the associated change in FLC clearance [12,13]. For the RIISC cohort, SPEP and immunofixation were also performed on baseline serum using standard laboratory procedures.

## Study design

Patients were recruited prospectively in all 3 cohorts, and data and biological samples collected at baseline visits were used for this analysis. Years of recruitment, end of follow-up, and median follow-up time for each study are given in Table 1. Time-to-event data were collected for 2 clinical endpoints: (1) kidney failure (defined as the initiation of KRT) and (2) death.

The following variables were available for analysis: age, sex, ethnicity (white, black, South Asian, or other), smoking status (current smoker, previous smoker, never smoked), co-morbidities (diabetes mellitus [DM], ischaemic heart disease [IHD], cerebrovascular disease, peripheral artery disease [PAD], chronic obstructive pulmonary disease [COPD], and malignancy), cause of CKD (vascular, diabetes, glomerular, tubulointerstitial, cystic or congenital, or other or unknown), mean arterial pressure (MAP), estimated glomerular filtration rate (eGFR, calculated using the 4-variable Modification of Diet in Renal Disease formula), and urine albumin-to-creatinine ratio (ACR).

**Table 1. Number of participants included and characteristics of each cohort study.** ACR in mg/mmol; eGFR in ml/min/1.73 m$^2$.

| Study | Number included | Setting | Inclusion criteria | Exclusion criteria | Years of recruitment | End of follow-up | Median (IQR) follow-up (months) |
|-------|-----------------|---------|--------------------|--------------------|----------------------|------------------|--------------------------------|
| RIISC | 878 | Secondary care | (1) eGFR < 30 or (2) eGFR 30–59 with (a) eGFR decline* or (b) urine ACR > 70 | (1) Previous dialysis or kidney transplant (2) Immunosuppression for immune-mediated kidney disease | 2010 to 2015 | End of 2018 | 74 (64 to 83) |
| SKS | 861 | Secondary care | eGFR >10 to <60 | Previous dialysis or kidney transplant | 2002 to 2010 | End of 2017 | 139 (110 to 161) |
| RRID | 1,739 | Primary care | eGFR 30–59 | (1) Expected survival <1 year (2) Previous solid organ transplant | 2008 to 2010 | End of 2015 | 61 (60 to 63) |

eGFR units are ml/min/1.73 m$^2$; ACR units are mg/mmol.

*eGFR decline defined as ≥ 5 ml/min/1.73 m$^2$ per year, or ≥ 10 ml/min/1.73 m$^2$ over 5 years.

ACR, albumin-to-creatinine ratio; eGFR, estimated glomerular filtration rate; IQR, interquartile range; RIISC, Renal Impairment in Secondary Care; RRID, Renal Risk in Derby; SKS, Salford Kidney Study.

No formal sample size calculations were carried out for these analyses, which were performed using the available specimen collections and datasets.

### Statistical analysis

Missing data were assumed to be missing at random, and multiple imputation using chained equations was performed. Results of complete case analyses were similar and are provided in S1 and S2 Tables.

Continuous variables all had skewed distributions as assessed by histograms. The relationships of MG or LC-MG status with other categorical baseline variables were assessed using Fisher's exact test, and relationships with continuous variables were assessed using the Wilcoxon rank-sum test.

The prognostic significance of an MG or LC-MG for risk of kidney failure was estimated by competing-risks regression (the Fine and Gray method [14]) to account for the competing risk of death, and expressed as a subhazard ratio (SHR) with 95% confidence interval (CI). The associations with risk of death were estimated using Cox proportional hazards models and expressed as hazard ratios (HRs) with 95% CI. Log–log plots were assessed for each variable to ensure that the proportional hazards assumption was not violated. Following an initial statistical review of our study, robust estimates of variance were used in all regression models, which are less sensitive to the presence of outliers. In analyses that included amalgamated data from all 3 cohorts, clustering was accounted for by the use of stratified models, in which the baseline hazard is allowed to vary by cohort. However, the estimated coefficient for a particular predictor variable is equal across cohorts.

All variables included in multivariable models were pre-specified, based on established prognostic factors for kidney failure and death in patients with CKD. Dummy variables were created for categorical variables with more than 2 categories, and fractional polynomials were used to explore the presence of non-linear relationships between continuous predictors and each outcome. Where fractional polynomials provided a better model fit, plots of risk against the variable on its original scale are presented to aid appreciation of the non-linear relationship.

## Results

### Any non-malignant MG

In total, 878 participants from the RIISC cohort were included, and 102 (11.6%) of these had an MG. Types of MG were as follows: 63 (61.8%) were IgG, 8 (7.8%) were IgM, 5 (4.9%) were IgA, 1 (1.0%) was biclonal (IgG and IgM), and 25 (24.5%) were LC-MG. Study population characteristics and the relationship between MG status and other baseline variables are shown in Table 2. Compared to those without an MG, those with an MG were on average older ($P < 0.001$), and a higher proportion had a history of malignancy ($P = 0.037$). For all other baseline variables, there were no statistically significant differences between those with and those without an MG.

### Kidney failure

In total, 327 (37.2%) participants progressed to kidney failure, with rates per 100 person-years of 10.5 and 9.3 for those with and without MG, respectively. The univariable associations between baseline variables and the risk of kidney failure are shown in Table 2. Age, eGFR, and urine ACR had non-linear relationships with the risk of kidney failure (Fig 1). The presence of

**Table 2. Baseline characteristics of 878 participants of the RIISC study with CKD by MG status.**

| Variable | All | With MG | Without MG | Completeness of data (%) |
|---|---|---|---|---|
| **N (%)** | 878 | 102 (11.6) | 776 (88.4) | 100 |
| **Age (years)** | 64.6 (51.7 to 76.0) | 73.8 (59.8 to 81.4) | 63.7 (50.2 to 75.5) | 100 |
| **Sex male** | 542 (61.7) | 66 (64.7) | 476 (61.3) | 100 |
| **Ethnicity** | | | | 100 |
| White | 598 (68.1) | 68 (66.7) | 530 (68.3) | |
| South Asian | 188 (21.4) | 24 (23.5) | 164 (21.1) | |
| Black | 84 (9.6) | 9 (8.9) | 75 (9.7) | |
| Other | 8 (0.9) | 1 (1.0) | 7 (0.9) | |
| **Co-morbidities** | | | | 100 |
| DM | 341 (38.8) | 48 (47.1) | 293 (37.8) | |
| IHD | 208 (23.7) | 30 (29.4) | 178 (22.9) | |
| Cerebrovascular disease | 102 (11.6) | 15 (14.7) | 87 (11.2) | |
| PAD | 93 (10.6) | 14 (13.7) | 79 (10.2) | |
| COPD | 89 (10.1) | 8 (7.8) | 81 (10.4) | |
| Malignancy | 128 (14.6) | 22 (21.6) | 106 (13.7) | |
| **Smoking status** | | | | 98.2 |
| Never | 416 (48.3) | 47 (47.0) | 369 (48.4) | |
| Previous | 333 (38.6) | 40 (40.0) | 293 (38.5) | |
| Current | 113 (13.1) | 13 (13.0) | 100 (13.1) | |
| **Cause of CKD** | | | | 91.2 |
| Vascular | 230 (28.7) | 34 (36.2) | 196 (27.7) | |
| Diabetes | 125 (15.6) | 20 (21.3) | 105 (14.9) | |
| Glomerular | 109 (13.6) | 7 (7.4) | 102 (14.4) | |
| Tubulointerstitial | 89 (11.1) | 6 (6.4) | 83 (11.7) | |
| Cystic or congenital | 66 (8.2) | 4 (4.3) | 62 (8.8) | |
| Other or unknown | 182 (22.7) | 23 (24.5) | 159 (22.5) | |
| **MAP (mm Hg)** | 93 (85 to 102) | 92 (83 to 103) | 93 (86 to 102) | 97.6 |
| **eGFR (ml/min/1.73 m$^2$)** | 31 (23 to 42) | 28 (22 to 42) | 31 (23 to 42) | 96.8 |
| **Urine ACR (mg/mmol)** | 33.4 (6.3 to 130.0) | 32.7 (5.6 to 161.2) | 33.4 (6.5 to 122.7) | 94.0 |

Categorical variables are shown as frequency (percentage), and continuous variables as median (interquartile range).

ACR, albumin-to-creatinine ratio; CKD, chronic kidney disease; COPD, chronic obstructive pulmonary disease; DM, diabetes mellitus; eGFR, estimated glomerular filtration rate; IHD, ischaemic heart disease; MAP, mean arterial pressure; MG, monoclonal gammopathy; PAD, peripheral artery disease; RIISC, Renal Impairment in Secondary Care.

an MG did not have a significant association with the risk of kidney failure (SHR 0.97 [95% CI 0.68 to 1.38], $P = 0.85$; Fig 2).

The multivariable model for kidney failure is shown in Table 2. After adjustment for age, sex, ethnicity, cause of CKD, MAP, eGFR, and urine ACR, the presence of an MG was not significantly associated with risk of kidney failure (SHR 1.16 [95% CI 0.80 to 1.69], $P = 0.43$). Younger age, female sex, black ethnicity, a cystic or congenital cause of CKD, a lower eGFR, and a higher urine ACR were all associated with a higher risk of kidney failure. In this multivariable model, age, eGFR, and urine ACR had non-linear associations with risk of kidney failure (Fig 3).

## Death

In total, 202 (23.0%) participants died. The death rates per 100 person-years were 10.8 and 5.3 for those with and without MG, respectively. The presence of an MG was associated with a

**Table 3. Association between baseline variables and risk of kidney failure (competing-risks regression, expressed as SHR with 95% CI) and death (Cox proportional hazards regression, expressed as HR with 95% CI) in 878 participants from the RIISC study with CKD.**

| Variable | Kidney failure | | | | | | Death | | | | | |
|---|---|---|---|---|---|---|---|---|---|---|---|---|
| | Univariable | | | Multivariable | | | Univariable | | | Multivariable | | |
| | SHR | 95% CI | *P* | SHR | 95% CI | *P* | HR | 95% CI | *P* | HR | 95% CI | *P* |
| **With MG** | 0.97 | 0.68 to 1.38 | 0.85 | 1.16 | 0.80 to 1.69 | 0.43 | 2.13 | 1.49 to 3.02 | <0.001 | 1.37 | 0.93 to 2.00 | 0.11 |
| **Age** | 1.00[a] | 1.00 to 1.00 | <0.001 | 1.00[a] | 0.99 to 1.00 | <0.001 | 3.36 | 2.73 to 4.12 | <0.001 | 2.83 | 2.21 to 3.64 | <0.001 |
| **Male sex** | 0.99 | 0.79 to 1.23 | 0.92 | 0.55 | 0.44 to 0.69 | <0.001 | 1.27 | 0.95 to 1.69 | 0.11 | 0.88 | 0.62 to 1.24 | 0.46 |
| **Ethnicity** | | | | | | | | | | | | |
| White | Ref | | | Ref | | | Ref | | | Ref | | |
| South Asian | 2.02 | 1.58 to 2.57 | <0.001 | 1.29 | 0.98 to 1.69 | 0.07 | 0.51 | 0.33 to 0.78 | 0.002 | 0.91 | 0.58 to 1.42 | 0.67 |
| Black | 1.98 | 1.42 to 2.76 | <0.001 | 1.77 | 1.32 to 2.38 | <0.001 | 0.80 | 0.48 to 1.33 | 0.39 | 1.13 | 0.67 to 1.90 | 0.65 |
| Other | 2.64 | 1.07 to 6.55 | 0.036 | 1.82 | 0.91 to 3.62 | 0.09 | 0.56 | 0.08 to 3.86 | 0.56 | 0.66 | 0.16 to 2.72 | 0.57 |
| **Co-morbidities** | | | | | | | | | | | | |
| DM | 0.92 | 0.73 to 1.15 | 0.46 | | | | 1.64 | 1.25 to 2.16 | <0.001 | 1.27 | 0.94 to 1.72 | 0.12 |
| IHD | 0.85 | 0.65 to 1.11 | 0.22 | | | | 2.44 | 1.83 to 3.24 | <0.001 | 1.44 | 1.05 to 1.96 | 0.022 |
| Cerebrovascular disease | 0.77 | 0.53 to 1.13 | 0.18 | | | | 1.97 | 1.38 to 2.81 | <0.001 | 1.27 | 0.87 to 1.85 | 0.21 |
| PAD | 0.86 | 0.59 to 1.27 | 0.45 | | | | 2.21 | 1.57 to 3.11 | <0.001 | 1.27 | 0.85 to 1.91 | 0.24 |
| COPD | 0.45 | 0.28 to 0.72 | 0.001 | | | | 1.46 | 0.99 to 2.16 | 0.06 | 1.14 | 0.74 to 1.77 | 0.55 |
| Malignancy | 0.51 | 0.35 to 0.76 | 0.001 | | | | 2.16 | 1.56 to 2.99 | <0.001 | 1.56 | 1.10 to 2.22 | 0.013 |
| **Smoking status** | | | | | | | | | | | | |
| Never | Ref | | | | | | Ref | | | Ref | | |
| Previous | 0.69 | 0.54 to 0.88 | 0.003 | | | | 1.73 | 1.28 to 2.34 | <0.001 | 1.06 | 0.76 to 1.49 | 0.74 |
| Current | 1.07 | 0.78 to 1.47 | 0.71 | | | | 1.14 | 0.71 to 1.84 | 0.58 | 1.25 | 0.70 to 2.24 | 0.45 |
| **Cause of CKD** | | | | | | | | | | | | |
| Vascular | Ref | | | Ref | | | Ref | | | | | |
| Diabetes | 1.92 | 1.33 to 2.78 | 0.001 | 1.05 | 0.69 to 1.60 | 0.81 | 0.81 | 0.52 to 1.26 | 0.35 | | | |
| Glomerular | 1.19 | 0.81 to 1.76 | 0.38 | 1.00 | 0.66 to 1.51 | 1.00 | 0.22 | 0.11 to 0.41 | <0.001 | | | |
| Tubulointerstitial | 0.89 | 0.57 to 1.38 | 0.59 | 0.63 | 0.37 to 1.06 | 0.08 | 0.31 | 0.16 to 0.59 | <0.001 | | | |
| Cystic or congenital | 2.85 | 2.01 to 4.04 | <0.001 | 3.99 | 2.74 to 5.83 | <0.001 | 0.26 | 0.10 to 0.63 | 0.003 | | | |
| Other or unknown | 1.24 | 0.89 to 1.73 | 0.21 | 1.21 | 0.85 to 1.73 | 0.28 | 0.82 | 0.57 to 1.17 | 0.27 | | | |
| **MAP** | 1.39 | 1.25 to 1.54 | <0.001 | 0.93 | 0.83 to 1.06 | 0.28 | 0.79 | 0.68 to 0.93 | 0.005 | | | |
| **eGFR** | | | | | | | | | | | | |
| | 1.18[b] | 1.14 to 1.22 | <0.001 | 0.94[b] | 0.93 to 0.96 | <0.001 | 0.45 | 0.36 to 0.56 | <0.001 | 0.67 | 0.53 to 0.86 | 0.002 |
| | 1.06[c] | 1.05 to 1.07 | <0.001 | 3.26[d] | 2.73 to 3.91 | <0.001 | | | | | | |
| **Urine ACR** | | | | | | | | | | | | |
| | 1.48[e] | 1.37 to 1.59 | <0.001 | 3.30[f] | 2.61 to 4.17 | <0.001 | 0.78[g] | 0.63 to 0.96 | 0.018 | 1.24 | 1.07 to 1.45 | 0.005 |
| | | | | 1.00[a] | 1.00 to 1.00 | <0.001 | 1.01[a] | 1.00 to 1.01 | <0.001 | | | |

Continuous variables are linear per 1 standard deviation unless otherwise indicated. Transformed continuous variables are indicated by the following footnotes: (a) $x^3$; (b) $x^{-2}$; (c) $x^{-2}\ln(x)$; (d) $x^{-1}$; (e) $\ln(x)$; (f) $x^{0.5}$; (g) $x$. Two rows for a continuous variable indicate the SHR or HR for each power of the degree-2 fractional polynomial transformation. Cells for variables not included in a particular multivariable model are shaded grey.

ACR, albumin-to-creatinine ratio; CI, confidence interval; CKD, chronic kidney disease; COPD, chronic obstructive pulmonary disease; DM, diabetes mellitus; eGFR, estimated glomerular filtration rate; HR, hazard ratio; IHD, ischaemic heart disease; MAP, mean arterial pressure; MG, monoclonal gammopathy; PAD, peripheral artery disease; RIISC, Renal Impairment in Secondary Care; SHR, subhazard ratio.

higher risk of death in the univariable model (HR 2.13 [95% CI 1.49 to 3.02], *P* < 0.001), as shown in Table 2 and Fig 4. However, in the multivariable model, adjusting for age, sex, ethnicity, co-morbidities, smoking status, eGFR, and urine ACR, the presence of an MG no longer had a statistically significant association with death (HR 1.37 [95% CI 0.93 to 2.00],

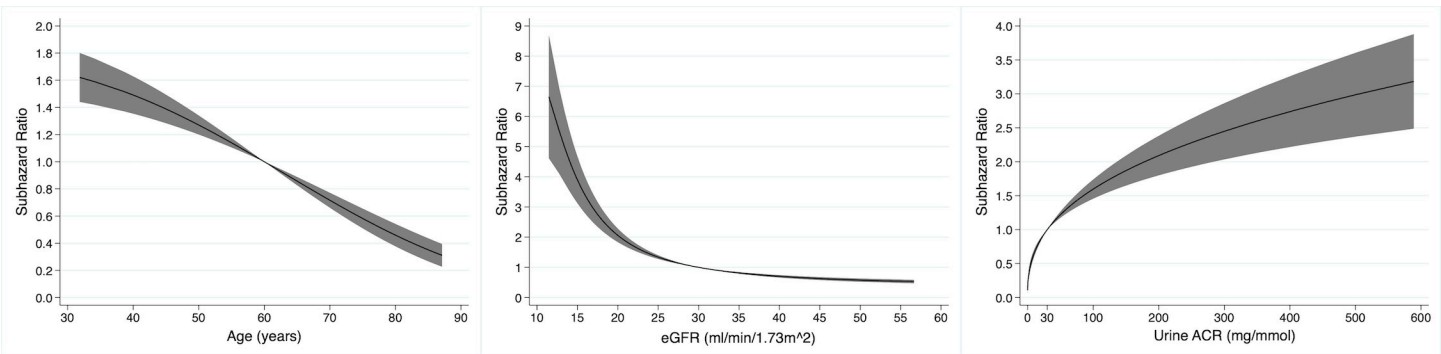

**Fig 1. Non-linear univariable associations with risk of kidney failure in 878 participants of the Renal Impairment in Secondary Care study with chronic kidney disease for age (relative to 60 years), estimated glomerular filtration rate (eGFR, relative to 30 ml/min/1.73 m²), and urine albumin-to-creatinine ratio (ACR, relative to 30 mg/mmol).**

$P$ = 0.11; Table 2). Older age, a history of IHD or malignancy, lower eGFR, and higher urine ACR were associated with a higher risk of death.

## Non-malignant LC-MG

In total, 3,478 participants from the 3 cohorts were included, and 55 (1.6%) of these had an LC-MG. Table 3 shows the study population characteristics and the relationship between

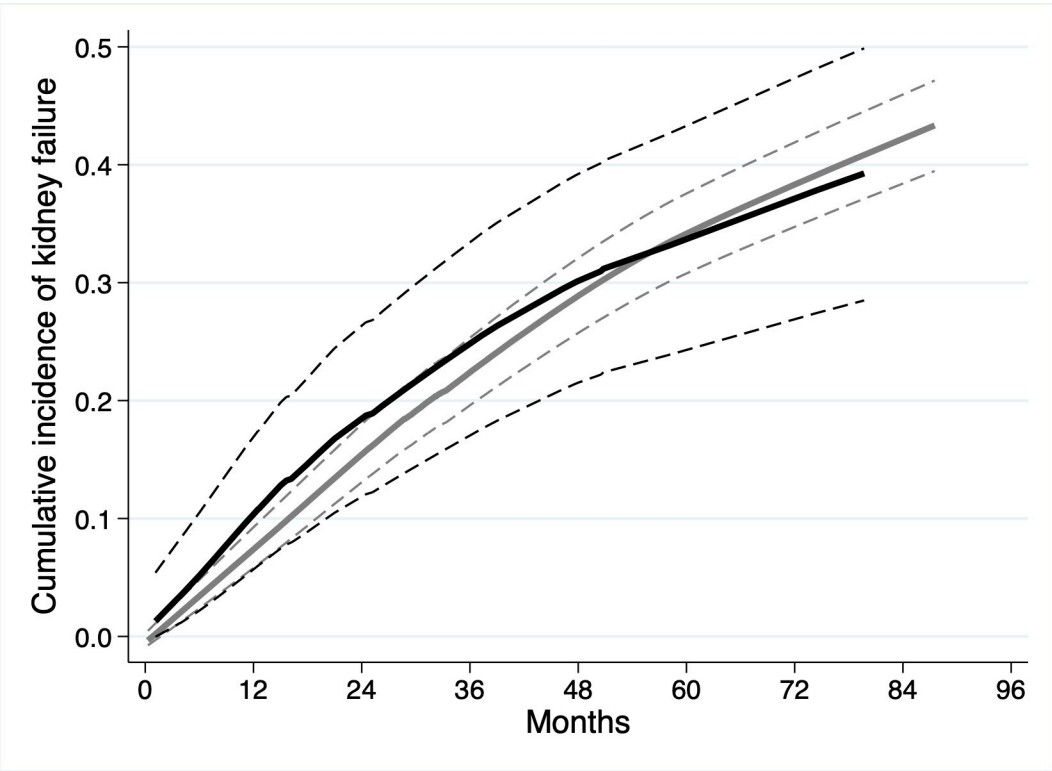

**Fig 2. Cumulative incidence function for kidney failure in 878 participants of the Renal Impairment in Secondary Care study with chronic kidney disease, by presence or absence of monoclonal gammopathy (MG).** MG present (black line); MG absent (grey line). Dashed lines represent 95% CIs.

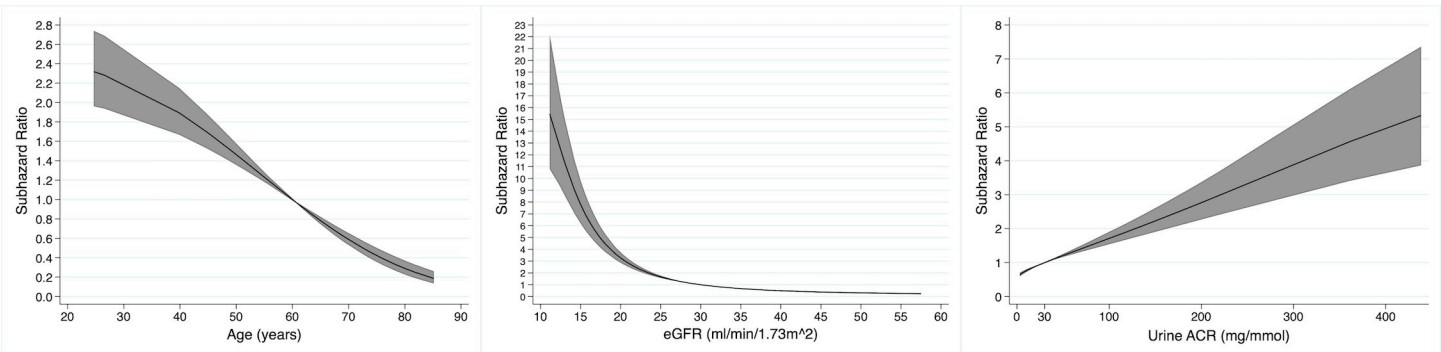

**Fig 3. Non-linear associations with risk of kidney failure in 878 participants of the Renal Impairment in Secondary Care study with chronic kidney disease (CKD) in a multivariable model containing monoclonal gammopathy status, age, sex, ethnicity, cause of CKD, mean arterial pressure, estimated glomerular filtration rate (eGFR), and urine albumin-to-creatinine ratio (ACR), for age (relative to 60 years), eGFR (relative to 30 ml/min/1.73 m$^2$), and urine ACR (relative to 30 mg/mmol).**

LC-MG status and other baseline variables. Compared to those without an LC-MG, those with an LC-MG were on average older ($P < 0.001$), a higher proportion were male ($P = 0.006$) and of black ethnicity ($P = 0.004$), and a lower proportion had a history of IHD ($P = 0.001$) or PAD ($P = 0.004$). There were no statistically significant differences between those with and those without LC-MG with respect to all other baseline variables.

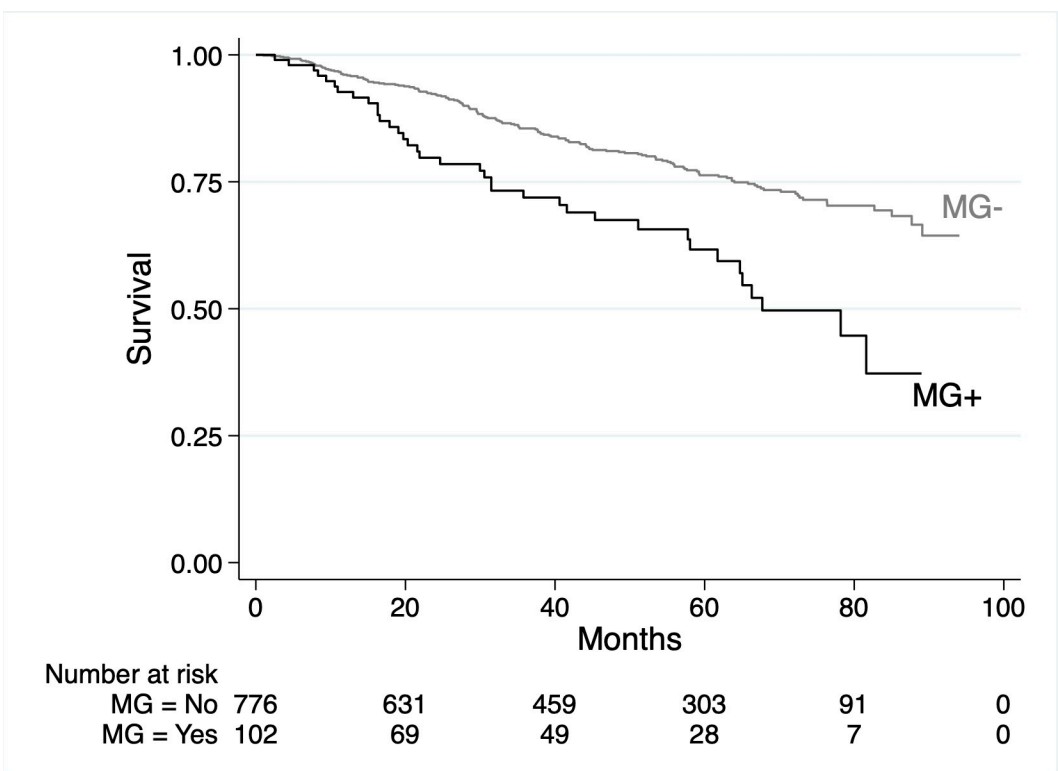

| Number at risk | | | | | | |
| --- | --- | --- | --- | --- | --- | --- |
| MG = No | 776 | 631 | 459 | 303 | 91 | 0 |
| MG = Yes | 102 | 69 | 49 | 28 | 7 | 0 |

**Fig 4. Kaplan–Meier survival curves for 878 participants of the Renal Impairment in Secondary Care study with chronic kidney disease by the presence or absence of monoclonal gammopathy (MG).** MG present (black line); MG absent (grey line).

**Table 4. Baseline characteristics for 3,478 participants with CKD from 3 cohort studies, by LC-MG status.**

| Variable | All | With LC-MG | Without LC-MG | Completeness of data (%) |
|---|---|---|---|---|
| **N (%)** | 3,478 | 55 (1.6) | 3,423 (98.4) | |
| **Age (years)** | 71.0 (61.2 to 78.0) | 77.8 (71.0 to 82.0) | 71.0 (61.1 to 78.0) | 100 |
| **Sex (male)** | 1,760 (50.6) | 38 (69.1) | 1,722 (50.3) | 100 |
| **Ethnicity** | | | | 100 |
| White | 3,126 (89.9) | 44 (80.0) | 3,082 (90.0) | |
| South Asian | 237 (6.8) | 5 (9.1) | 232 (6.8) | |
| Black | 96 (2.8) | 6 (10.9) | 90 (2.6) | |
| Other | 19 (0.6) | 0 (0.0) | 19 (0.6) | |
| **Co-morbidities** | | | | 100 |
| DM | 914 (26.3) | 16 (29.1) | 898 (26.2) | |
| IHD | 1,347 (38.7) | 10 (18.2) | 1,337 (39.1) | |
| Cerebrovascular disease | 395 (11.4) | 3 (5.5) | 392 (11.5) | |
| PAD | 879 (25.3) | 5 (9.1) | 874 (25.5) | |
| **Smoking status** | | | | 99.5 |
| Never | 1,486 (43.0) | 26 (47.3) | 1,460 (42.9) | |
| Previous | 1,667 (48.2) | 28 (50.9) | 1,639 (48.1) | |
| Current | 307 (8.9) | 1 (1.8) | 306 (9.0) | |
| **MAP (mm Hg)** | 93 (86 to 102) | 92 (85 to 99) | 93 (86 to 102) | 99.3 |
| **eGFR (ml/min/1.73 m$^2$)** | 42.3 (26.2 to 54.4) | 40.4 (24.3 to 54.2) | 42.3 (26.3 to 54.4) | 99.2 |
| **Urine ACR (mg/mmol)** | 3.4 (0.3 to 27.3) | 4.7 (0.4 to 76.6) | 3.4 (0.2 to 26.7) | 95.5 |

Categorical variables are shown as frequency (percentage), and continuous variables as median (interquartile range).

ACR, albumin-to-creatinine ratio; CKD, chronic kidney disease; DM, diabetes mellitus; eGFR, estimated glomerular filtration rate; IHD, ischaemic heart disease; LC-MG, light chain monoclonal gammopathy; MAP, mean arterial pressure; PAD, peripheral artery disease.

## Kidney failure

In total, 564 (16.2%) patients progressed to kidney failure, with rates per 100 person-years of 4.9 and 3.2 for those with and without an LC-MG, respectively. The univariable associations between baseline variables and the risk of kidney failure are shown in Table 5. Age, eGFR, and urine ACR had non-linear associations with risk of kidney failure in the univariable analyses (Fig 5). The presence of an LC-MG did not have a significant association with the risk of kidney failure (SHR 1.07 [95% CI 0.58 to 1.96], P = 0.82; Fig 6).

The multivariable model for kidney failure is shown in Table 5. After adjusting for age, sex, ethnicity, MAP, eGFR, and urine ACR, the presence of an LC-MG did not have a statistically significant association with risk of kidney failure (SHR 1.42 [95% CI 0.78 to 2.57], P = 0.26). In this multivariable model, a younger age, black ethnicity, a lower eGFR, and a higher urine ACR were associated with a higher risk of kidney failure, and the non-linear associations with age, eGFR, and urine ACR are shown in Fig 7.

## Death

In total, 803 (23.1%) participants died. Death rates were 9.3 and 4.5 per 100 person-years for those with and without an LC-MG, respectively. The univariable associations between baseline factors and death are shown in Table 5. LC-MG was associated with a higher risk of death (HR 2.51 [95% CI 1.59 to 3.96], P < 0.001), and Fig 8 shows Kaplan–Meier survival curves by LC-MG status. The univariable analyses showed that MAP, eGFR, and urine ACR had non-linear associations with risk of death (Fig 9). In the multivariable model (Table 5), after adjustment

**Table 5. Association between baseline variables and risk of kidney failure (competing-risks regression, expressed as SHR with 95% CI) and death (Cox proportional hazards regression, expressed as HR with 95% CI) in 3,478 participants with CKD from 3 cohort studies.**

| Variable | Kidney failure | | | | | | Death | | | | | |
|---|---|---|---|---|---|---|---|---|---|---|---|---|
| | Univariable | | | Multivariable | | | Univariable | | | Multivariable | | |
| | SHR | 95% CI | *P* | SHR | 95% CI | *P* | HR | 95% CI | *P* | HR | 95% CI | *P* |
| **With LC-MG** | 1.07 | 0.58 to 1.96 | 0.82 | 1.42 | 0.78 to 2.57 | 0.26 | 2.51 | 1.59 to 3.96 | <0.001 | 1.49 | 0.93 to 2.39 | 0.10 |
| **Age** | | | | | | | | | | | | |
| | 1.01[a] | 1.00 to 1.02 | 0.20 | 1.00[a] | 1.00 to 1.00 | <0.001 | 2.88 | 2.60 to 3.19 | <0.001 | 2.76 | 2.48 to 3.08 | <0.001 |
| | 1.00[b] | 0.99 to 1.00 | 0.050 | | | | | | | | | |
| **Male sex** | 0.95 | 0.81 to 1.12 | 0.53 | 1.14 | 0.96 to 1.37 | 0.13 | 1.59 | 1.37 to 1.84 | <0.001 | 1.27 | 1.09 to 1.49 | 0.002 |
| **Ethnicity** | | | | | | | | | | | | |
| White | Ref | | | Ref | | | Ref | | | Ref | | |
| South Asian | 1.94 | 1.56 to 2.41 | <0.001 | 1.15 | 0.89 to 1.48 | 0.30 | 0.67 | 0.47 to 0.94 | 0.022 | 1.11 | 0.79 to 1.56 | 0.56 |
| Black | 1.84 | 1.35 to 2.49 | <0.001 | 1.71 | 1.29 to 2.27 | <0.001 | 0.77 | 0.48 to 1.25 | 0.30 | 1.10 | 0.70 to 1.73 | 0.69 |
| Other | 2.80 | 1.34 to 5.86 | 0.006 | 1.42 | 0.55 to 3.62 | 0.47 | 0.48 | 0.12 to 1.94 | 0.30 | 0.79 | 0.29 to 2.14 | 0.64 |
| **Co-morbidities** | | | | | | | | | | | | |
| DM | 0.92 | 0.77 to 1.09 | 0.31 | | | | 1.71 | 1.48 to 1.97 | <0.001 | 1.42 | 1.22 to 1.65 | <0.001 |
| IHD | 1.03 | 0.85 to 1.24 | 0.78 | | | | 1.64 | 1.35 to 1.99 | <0.001 | 1.31 | 1.10 to 1.56 | 0.002 |
| Cerebrovascular disease | 0.84 | 0.64 to 1.10 | 0.20 | | | | 1.97 | 1.65 to 2.36 | <0.001 | 1.39 | 1.15 to 1.69 | 0.001 |
| PAD | 1.10 | 0.88 to 1.36 | 0.41 | | | | 0.93 | 0.70 to 1.22 | 0.58 | 0.93 | 0.76 to 1.14 | 0.49 |
| **Smoking status** | | | | | | | | | | | | |
| Never | Ref | | | | | | Ref | | | Ref | | |
| Previous | 0.72 | 0.60 to 0.86 | <0.001 | | | | 1.71 | 1.47 to 2.00 | <0.001 | 1.26 | 1.07 to 1.48 | 0.005 |
| Current | 1.14 | 0.91 to 1.44 | 0.26 | | | | 1.34 | 1.04 to 1.74 | 0.026 | 1.56 | 1.16 to 2.09 | 0.003 |
| **MAP** | | | | | | | | | | | | |
| | 1.33 | 1.23 to 1.44 | <0.001 | 1.07 | 0.97 to 1.18 | 0.16 | 0.00[c] | 0.00 to 0.00 | <0.001 | | | |
| | | | | | | | 71.39[d] | 6.10 to 835.26 | 0.001 | | | |
| **eGFR** | | | | | | | | | | | | |
| | 1.17[e] | 1.15 to 1.20 | <0.001 | 0.96[e] | 0.95 to 0.97 | <0.001 | 0.00[f] | 0.00 to 0.01 | <0.001 | 0.65 | 0.58 to 0.74 | <0.001 |
| | 1.06[g] | 1.05 to 1.06 | <0.001 | 2.4[h] | 2.10 to 2.80 | <0.001 | | | | | | |
| **Urine ACR** | | | | | | | | | | | | |
| | 10.97[i] | 7.20 to 16.70 | <0.001 | 3.58[i] | 2.62 to 4.90 | <0.001 | 1.68[i] | 1.18 to 2.40 | 0.005 | 1.15[c] | 1.09 to 1.22 | <0.001 |
| | 0.80[f] | 0.68 to 0.95 | 0.008 | | | | 0.40[j] | 0.29 to 0.55 | <0.001 | 1.01[d] | 1.00 to 1.01 | <0.001 |

Continuous variables are linear per 1 standard deviation unless otherwise indicated. Transformed continuous variables are indicated by the following footnotes: (a) $x^3$; (b) $x^3\ln(x)$; (c) $\ln(x)$; (d) $[\ln(x)]^2$; (e) $x^{-2}$; (f) $x^2$; (g) $x^{-2}\ln(x)$; (h) $x^{-1}$; (i) $x^{0.5}$; (j) $x^{0.5}\ln(x)$. Two rows for a continuous variable indicate the SHR or HR for each power of the degree-2 fractional polynomial transformation. Cells for variables not included in a particular multivariable model are shaded grey.

ACR, albumin-to-creatinine ratio; CI, confidence interval; CKD, chronic kidney disease; DM, diabetes mellitus; eGFR, estimated glomerular filtration rate; HR, hazard ratio; IHD, ischaemic heart disease; LC-MG, light chain monoclonal gammopathy; MAP, mean arterial pressure; PAD, peripheral artery disease; SHR, subhazard ratio.

for age, sex, ethnicity, co-morbidities, smoking status, eGFR, and urine ACR, the presence of an LC-MG did not have a statistically significant association with death (HR 1.49 [95% CI 0.93 to 2.39], *P* = 0.10). In this multivariable model, older age; male sex; a history of DM, IHD, or cerebrovascular disease; being a previous or current smoker, a lower eGFR, and a higher urine ACR (non-linear association; Fig 10) were associated with a higher risk of death.

## Discussion

We evaluated the prognostic significance of non-malignant MG in patients with CKD. We found the prevalence of non-malignant MG to be higher in our CKD population compared to

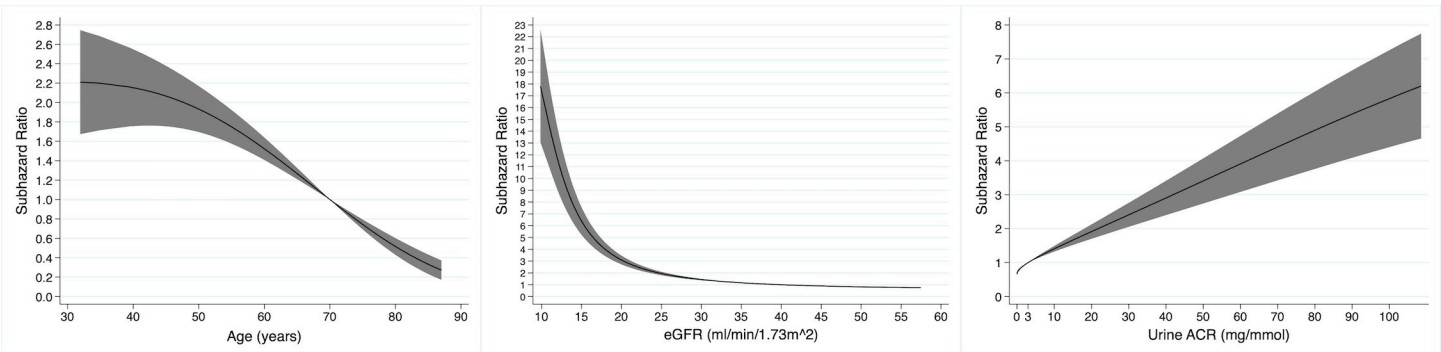

**Fig 5. Non-linear univariable associations with risk of kidney failure in 3,478 participants with chronic kidney disease from 3 cohort studies for age (relative to 70 years), estimated glomerular filtration rate (eGFR, relative to 40 ml/min/1.73 m², and urine albumin-to-creatinine ratio (ACR, relative to 3 mg/mmol).**

reported estimates of prevalence for the general population [1]. However, the presence of an MG was not associated with a higher risk of kidney failure or death after accounting for established prognostic factors in CKD.

To the best of our knowledge, only 1 other study, by Haynes et al. [7], has assessed the relationship between MG and clinical outcomes in patients with CKD. Compared to our study, that by Haynes et al. had far fewer participants (382), such that only 35 patients had MGUS. Therefore, there is likely to be less bias in our estimates of the risk of kidney failure or death associated with an MG in CKD.

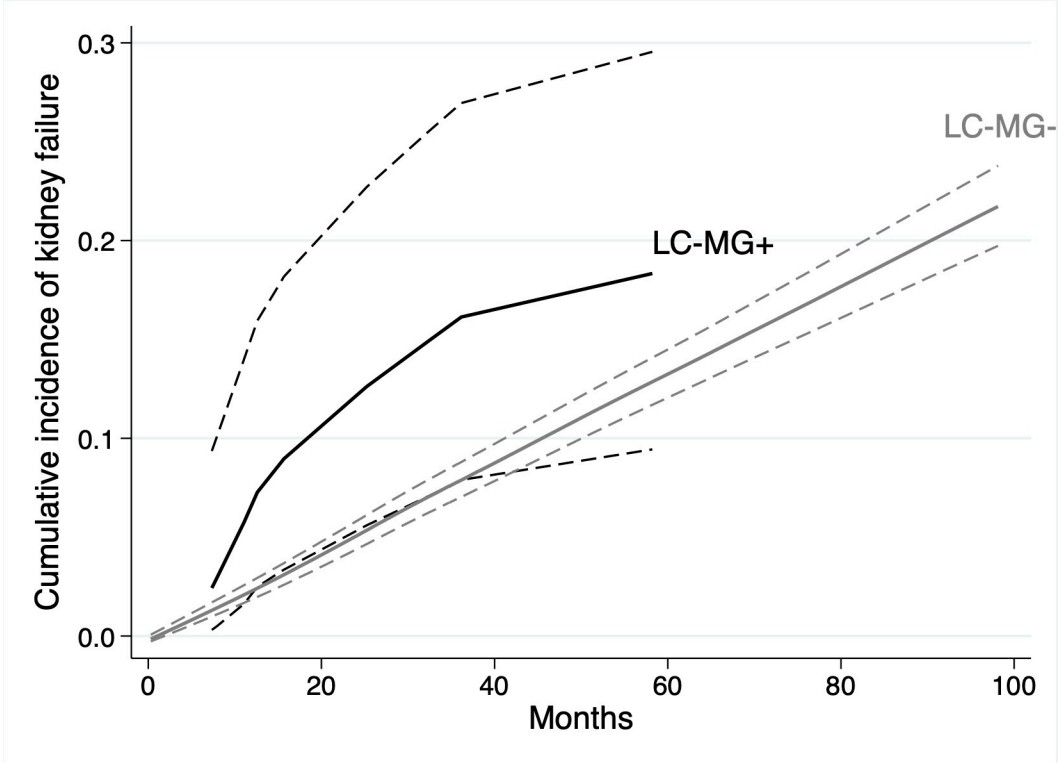

**Fig 6. Cumulative incidence function for kidney failure in 3,478 participants with chronic kidney disease from 3 cohort studies, by the presence or absence of light chain monoclonal gammopathy (LC-MG).** LC-MG present (black line); LC-MG absent (grey line). Dashed lines represent 95% CIs.

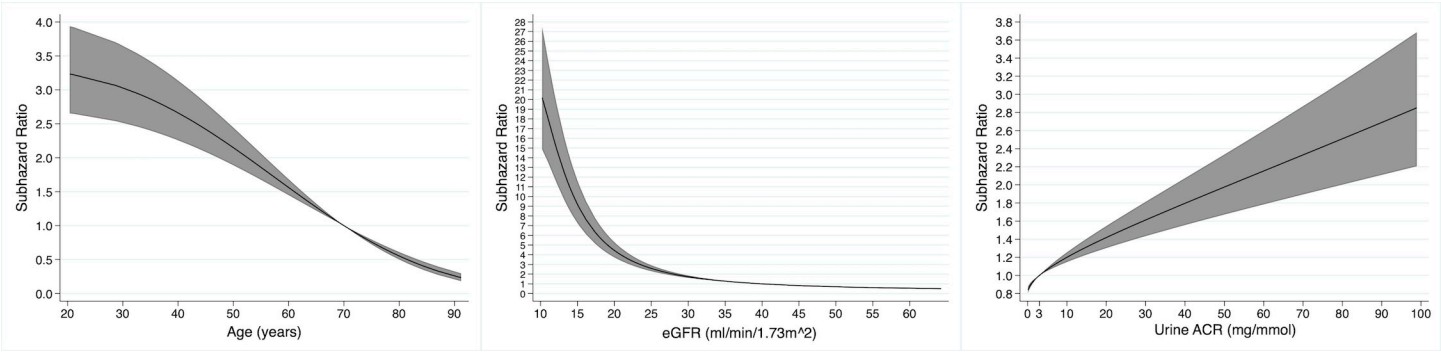

**Fig 7. Non-linear associations with risk of kidney failure in 3,478 participants with chronic kidney disease from 3 cohort studies in a multivariable model containing light chain monoclonal gammopathy status, age, sex, ethnicity, mean arterial pressure, estimated glomerular filtration rate (eGFR), and urine albumin-to-creatinine ratio (ACR), for age (relative to 70 years), eGFR (relative to 40 ml/min/1.73 m$^2$), and urine ACR (relative to 3 mg/mmol).**

The results of our study and the study by Haynes et al. suggest that the shorter survival associated with MGUS in the general population is not seen in patients with CKD. It is possible that neither study was large enough to detect a small increase in the risk of death. However, it may be that the already significantly increased rate of death in individuals with CKD compared to individuals without CKD renders any risk associated with an MG, which is primarily due to malignant transformation over time, negligible.

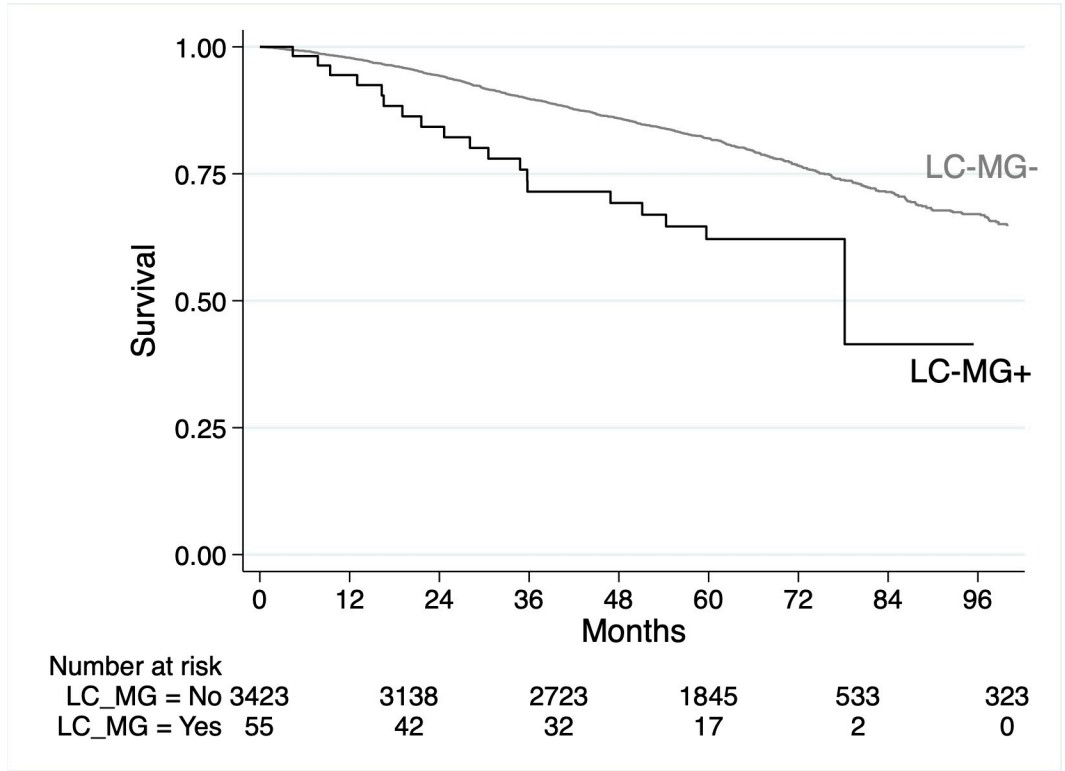

**Fig 8. Kaplan–Meier survival curves for 3,478 participants with chronic kidney disease from 3 cohort studies by the presence or absence of light chain monoclonal gammopathy (LC-MG).** LC-MG present (black line); LC-MG absent (grey line).

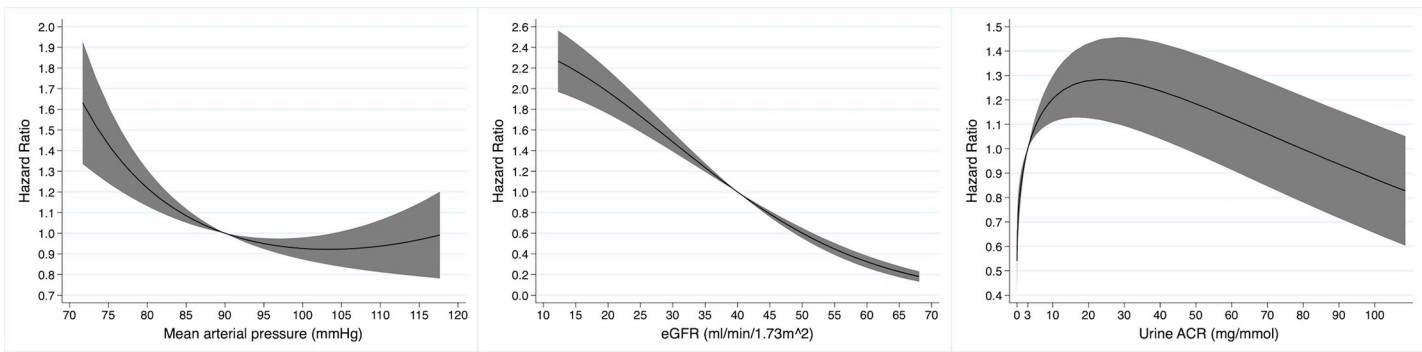

**Fig 9. Non-linear univariable associations with risk of death in 3,478 participants with chronic kidney disease from 3 cohort studies for mean arterial pressure (relative to 90 mm Hg), estimated glomerular filtration rate (eGFR, relative to 40 ml/min/1.73 m$^2$), and urine albumin-to-creatinine ratio (ACR, relative to 3 mg/mmol).**

Our finding of a higher prevalence of MG in CKD is also consistent with the study by Haynes et al. and appears attributable to an increased prevalence of both intact immunoglobulin MG and LC-MG. The prevalence of total MGUS in the Olmsted County cohort in individuals aged 70–79 was 5.9%, and the prevalence of light chain MGUS was 1.1% [15]; in our study, the crude prevalence was 11.6% for total MGUS (median age 73.8 years) and 1.6% for light chain MGUS (median age 77.8 years).

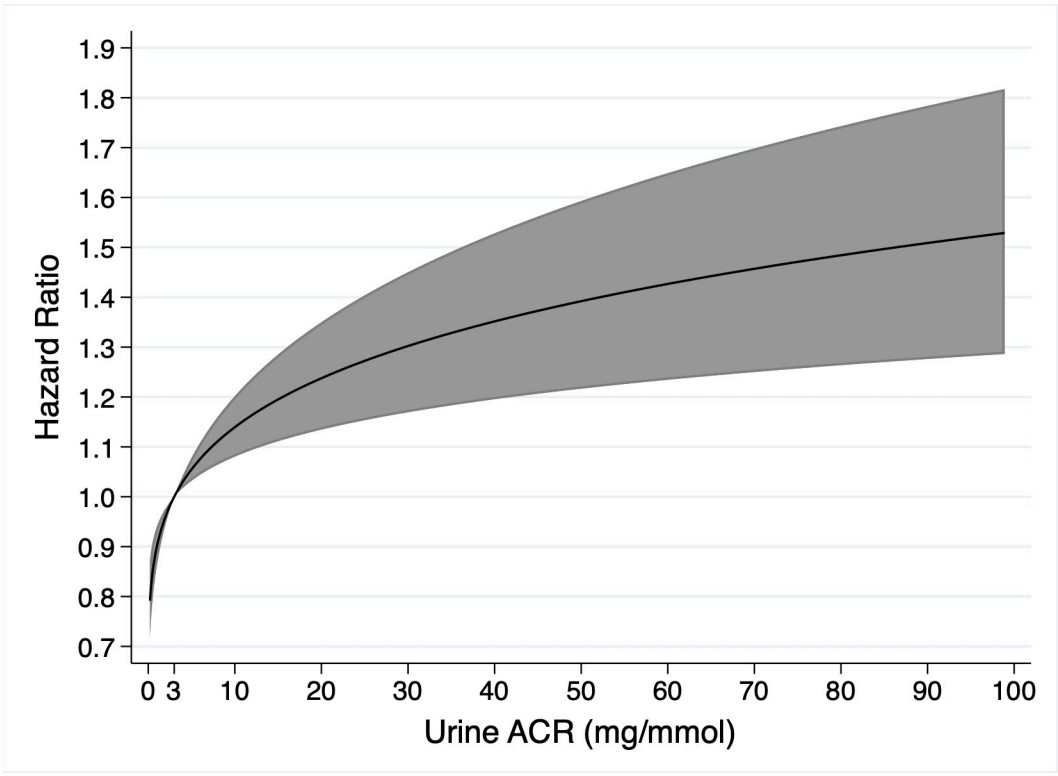

**Fig 10. Non-linear association of urine albumin-to-creatinine ratio (ACR, relative to 3 mg/mmol) with risk of death in 3,478 participants with chronic kidney disease from 3 cohort studies in a multivariable model containing light chain monoclonal gammopathy status, age, sex, ethnicity, co-morbidities, smoking status, estimated glomerular filtration rate, and urine ACR.**

A significant strength of this study was the inclusion of participants from multiple cohorts from both primary and secondary care and that it is the largest cohort to date of patients with MGUS and CKD. A significant limitation was the absence of SPEP and immunofixation data from the SKS and RRID cohorts. We could, therefore, detect only LC-MG in these cohorts, and many patients with an intact Ig MG would not have been identified. However, in the RIISC study, where SPEP and immunofixation were performed on serum from all participants, the presence of any non-malignant MG was not associated with a higher risk of kidney failure or death.

Further, we did not assess the association of MGUS in this CKD population with other clinically important outcomes that are associated with MGUS in those without CKD, such as cardiovascular events, infections, or the evolution of an MG to multiple myeloma or other paraprotein-related diseases. We also did not examine CKD progression (via change in eGFR) with time, which would be a more sensitive marker for MG-associated kidney damage than the outcome of kidney failure.

Further research is required concerning the prognostic implications of non-malignant MG in patients with CKD. However, based on the results of our study, these patients and their healthcare providers may be reassured that the presence of an MG does not significantly increase the risk of kidney failure or death that is associated with CKD.

In conclusion, the prevalence of non-malignant MG appears to be higher in patients with CKD than in the general population. However, its presence is not independently associated with a significantly higher risk of kidney failure or death.

## Supporting information

**S1 STROBE Checklist. The relevant section and paragraph of the paper is indicated for each item on the checklist.**
(DOCX)

**S1 Protocol. The prospectively written analysis plan for the study.**
(DOCX)

**S1 Table. Complete case analyses of the association between baseline variables and risk of kidney failure (competing-risks regression, expressed as SHR with 95% CI) and death (Cox proportional hazards regression, expressed as HR with 95% CI) in 878 participants from the RIISC study with CKD.** Continuous variables are linear per 1 standard deviation unless otherwise indicated. Transformed continuous variables are indicated by the following footnotes: (a) $x^3$; (b) $x^{-2}$; (c) $x^{-2}\ln(x)$; (d) $\ln(x)$; (e) $x^{-1}$; (f) $x^{0.5}$. ACR, albumin-to-creatinine ratio; CI, confidence interval; CKD, chronic kidney disease; COPD, chronic obstructive pulmonary disease; DM, diabetes mellitus; eGFR, estimated glomerular filtration rate; HR, hazard ratio; IHD, ischaemic heart disease; MAP, mean arterial pressure; MG, monoclonal gammopathy; PAD, peripheral artery disease; RIISC, Renal Impairment in Secondary Care; SHR, subhazard ratio.
(DOCX)

**S2 Table. Complete case analyses of the association between baseline variables and risk of kidney failure (competing-risks regression, expressed as SHR with 95% CI) and death (Cox proportional hazards regression, expressed as HR with 95% CI) in 3,478 participants with CKD from 3 cohort studies.** Continuous variables are linear per 1 standard deviation unless otherwise indicated. Transformed continuous variables are indicated by the following footnotes: (a) $x^3$; (b) $x^{-2}$; (c) $x^{-2}\ln(x)$; (d) $x^{0.5}$; (e) $x^2$; (f) $x^{-1}$; (g) $\ln(x)$; (h) $[\ln(x)]^2$; (i) $x^{0.5}\ln(x)$. ACR, albumin-to-creatinine ratio; CI, confidence interval; CKD, chronic kidney disease; DM,

diabetes mellitus; eGFR, estimated glomerular filtration rate; HR, hazard ratio; IHD, ischaemic heart disease; LC-MG, light chain monoclonal gammopathy; MAP, mean arterial pressure; PAD, peripheral artery disease; SHR, subhazard ratio.
(DOCX)

## Author Contributions

**Conceptualization:** Anthony Fenton, Paul Cockwell.

**Data curation:** Anthony Fenton, Rajkumar Chinnadurai, Latha Gullapudi, Petros Kampanis, Indranil Dasgupta, James Ritchie, Stephen Harding, Charles J. Ferro, Philip A. Kalra, Maarten W. Taal, Paul Cockwell.

**Formal analysis:** Anthony Fenton.

**Investigation:** Anthony Fenton, Petros Kampanis, Indranil Dasgupta, Stephen Harding, Philip A. Kalra, Maarten W. Taal, Paul Cockwell.

**Methodology:** Anthony Fenton, Charles J. Ferro, Philip A. Kalra, Maarten W. Taal, Paul Cockwell.

**Project administration:** Anthony Fenton, Rajkumar Chinnadurai, Latha Gullapudi, Indranil Dasgupta, Stephen Harding, Charles J. Ferro, Maarten W. Taal, Paul Cockwell.

**Resources:** Indranil Dasgupta, Stephen Harding, Charles J. Ferro, Philip A. Kalra, Maarten W. Taal, Paul Cockwell.

**Supervision:** Indranil Dasgupta, Charles J. Ferro, Philip A. Kalra, Maarten W. Taal, Paul Cockwell.

**Writing – original draft:** Anthony Fenton, Paul Cockwell.

**Writing – review & editing:** Rajkumar Chinnadurai, Latha Gullapudi, Petros Kampanis, Indranil Dasgupta, James Ritchie, Stephen Harding, Charles J. Ferro, Philip A. Kalra, Maarten W. Taal, Paul Cockwell.

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
