## [Decision Letter · Decision Letter 0]

15 Nov 2019

Dear Dr. Fenton,

Thank you very much for submitting your manuscript "Association between non-malignant monoclonal gammopathy and adverse outcomes in chronic kidney disease" (PMEDICINE-D-19-03271) for consideration at PLOS Medicine. 

[LINK]

In light of these reviews, I am afraid that we will not be able to accept the manuscript for publication in the journal in its current form, but we would like to consider a revised version that addresses the reviewers' and editors' comments. Obviously we cannot make any decision about publication until we have seen the revised manuscript and your response, and we plan to seek re-review by one or more of the reviewers. 

We expect to receive your revised manuscript by Dec 05 2019 11:59PM. Please email us (plosmedicine@plos.org) if you have any questions or concerns.

We look forward to receiving your revised manuscript. 

Sincerely,

Adya Misra, PhD

Senior Editor 

PLOS Medicine

plosmedicine.org

Title: please include a study descriptor to adhere to PLOS Medicine style. our title must be nondeclarative and not a question. It should begin with main concept if possible. "Effect of" should be used only if causality can be inferred, i.e., for an RCT. Please place the study design ("A randomized controlled trial," "A retrospective study," "A modelling study," etc.) in the subtitle (ie, after a colon).

Abstract- please clarify if a causal association can be inferred from population studies? Please consider toning down or removing

Abstract-please provide population demographics, ie where the cohorts are from including names of hospitals or countries as appropriate

Abstract Please introduce FLC ratio on first view

Abstract background- please rephrase the first sentence to “In studies including the general population..” or similar for clarity

Abstract methods and findings-please include a limitation of your work as the last sentence of this section

Abstract methods section requires clarification. Please simplify the text in the abstract to explain the FLC ratio/LC MG measurement. In line 31-33 you say “Further, to assess the

32 association between a light-chain (LC) MG (defined as an FLC ratio outside the renal

33 reference range with an increased level of the involved light chain) two other cohorts were

34 also studied…”. Could you clarify what association is being referred to here?

At this stage, we ask that you include a short, non-technical Author Summary of your research to make findings accessible to a wide audience that includes both scientists and non-scientists. The Author Summary should immediately follow the Abstract in your revised manuscript. This text is subject to editorial change and should be distinct from the scientific abstract. Please

see our author guidelines for more information: https://journals.plos.org/plosmedicine/s/revising-your-manuscript#loc-author-summary

Did your study have a prospective protocol or analysis plan? Please state this (either way) early in the Methods section. a) If a prospective analysis plan (from your funding proposal, IRB or other ethics committee submission, study protocol, or other planning document written before analyzing the data) was used in designing the study, please include the relevant prospectively written document with your revised manuscript as a Supporting Information file to be published alongside your study, and cite it in the Methods section. A legend for this file should be included at the end of your manuscript. b) If no such document exists, please make sure that the Methods section transparently describes when analyses were planned, and when/why any data-driven changes to analyses took place. c) In either case, changes in the analysis—including those made in response to peer review comments—should be identified as such in the Methods section of the paper, with rationale.

For all observational studies, in the manuscript text, please indicate: (1) the specific hypotheses you intended to test, (2) the analytical methods by which you planned to test them, (3) the analyses you actually performed, and (4) when reported analyses differ from those that were planned, transparent explanations for differences that affect the reliability of the study's results. If a reported analysis was performed based on an interesting but unanticipated pattern in the data, please be clear that the analysis was data-driven.

Line 100- Please introduce MGRS at first view

Please provide p values along with 95% CI where appropriate

Lines 164-165 require clarification- please can you explain what is meant by “more had a history of …” etc? 

Please ensure that the study is reported according to the STROBE guideline, and include the completed STROBE checklist as Supporting Information. Please add the following statement, or similar, to the Methods: "This study is reported as per the Strengthening the Reporting of Observational Studies in Epidemiology (STROBE) guideline (S1 Checklist)."

Please present and organize the Discussion as follows: a short, clear summary of the article's findings; what the study adds to existing research and where and why the results may differ from previous research; strengths and limitations of the study; implications and next steps for research, clinical practice, and/or public policy; one-paragraph conclusion.

Comments from the reviewers:

Reviewer #1: This is a retrospective analysis of data collected prospectively for other causes from 3 different sources, used to analyze if there is a correlation between CKD progression and MG (prognostic significance of MGUS in CKD patients). Hypotheses are well defined by the authors. 

First, as they are using data collected for other purposes, adding many variables to the statistical analysis limits the researchers' ability to determine cause and effect. It is difficult to assess correlation using a retrospective study (RS) design, as RS are subject to numerous threats to validity (limiting interpretation and generalizability of results): the results should be considered as preliminary. Thus, the authors should be very cautious in their conclusions.

Also, data collection comes from 3 differents sources, with different inclusion criteria, data collected, etc... Define if MG was detected by serum protein electrophoresis, if then patients underwent serum immunosubtraction and/or immunofixation (Immunofixation of a 24 h urine specimen performed to test for Bence Jones protein?). The diagnostic criteria for MGUS also should be included in this article and not referred for readers to the 3 studies used for it (these should all be included in the article itself): it would be great if authors could clarify eGFR formula used (CKD-EPI, MDRD 4...) and include the definition of MG/MGUS. Specify type of Ig related MG if possible.

There is a low proportion of events (1.6% including all 3 studies, 55 patients from 3478), which limits statistical power to detect associations: authors should write about their study SUGGESTING X or Y are/aren't risk factors, but that further prospective studies are neccessary to clarify. 

It would be interesting to know some serologies in these patients, as MG has been related to them (herpes virus, EBV, etc): Babel N, Schwarzmann F, Pruss A, et al. Monoclonal gammopathy of undetermined significance (MGUS) is associated with an increased frequency of Epstein‐Barr Virus (EBV) - Latently infected B lymphocytes in long‐term renal transplant patients. Transplant Proc 2004; 36: 2679; Regamey N, Hess V, Passweg J, et al. Infection with human herpesvirus 8 and transplant‐associated gammopathy. Transplantation 2004; 77: 1551.

Include Monoclonal gammopathy after liver transplantation: a risk factor for long‐term medical complications other than malignancies (DOI 10.1111/j.1432-2277.2011.01362.x), as they refer to CKD and MG (although in liver Tx recipients).

Overall, the results indeed suggested that there might be no association between CKD and MG, but no clear cut conclusion should be drawn from this (I would change their strong conclusion phrasing).

I believe it is an interesting topic for nephrologists, as nowadays the number of patients with MG that reach our clinics is growing.

Reviewer #2: I confine my remarks to statistical aspects of this paper. The general approach is fine, but I do have some issues to resolve before I can recommend publication.

One general thing is that I would move the supplemental figures into the text.

Line 55-56: This is too strong. This accepts the null. If you want to make this statement, you would have to do tests of equivalence. 

Line 130-131: In general, this is not a reason to not do power analysis. If the authors had done power analysis and found that they needed a larger N, then they would have had to either find more data or do a different analysis. In this specific case, since N is so large, it's not a problem.

Line 137-140 How many extreme points were there? Also, why not use robust Cox regression? 

Line 141-143 Why was this done? Here, statistical significance is not the issue. Also, the authors don't seem to use the results of these analyses (unless I missed something)

Line 148-149 Good!

Line 150 "Stratified" usually means "separate" I'm not sure that this is what is meant here. Was each cohort analyzed separately? It doesn't look like that was done, from the results. Please clarify. (An alternative would be a multilevel model, if the authors are worried about dependent errors).

Line 154: Good

Tables: Why are some sections blank?

Peter Flom

Reviewer #3: Fenton et al. reported the association between MGUS and CKD and found that no association with the progression of CKD. The statistical analyses wee well performed. However, it is difficult to find the role of MGUS for the progression of CKD. The presented data is very limited and it is very difficult to understand the the meaning of the statistical analyses. 

 Among MGUS, amyloidosis is a rare but an important disease for the progression of CKD. In general, patients with amyloidosis has large amount of urinary protein excretion witch has some association with the progression of CKD. However, there is no description about amyloidosis. If the authors investigate the association between amyloidosis and the progression of CKD, the results would be useful for nephrologist. Just the definition of MGUS is too wide to consider the role for the progression of CKD.

 There is no information on CRP. Since the infection has a key role for the progression of MGUS, the lack of such information should be avoided. 

 The data are collected at baseline visits. Although the authors described higher HR of ACR, urinary protein excretion may increase with time. So, it may be inadequate to investigate the role of urinary protein excretion by the baseline value. 

Reviewer #4: I have several comments:

* In addition to renal transplantation or dialysis, do the authors have data to consider other endpoints, such as acute kidney injury, or rapid CKD progression?

* The confidence intervals in the abstract are quite wide, so it is not inconceivable that the study did not have enough power to detect significance at the conventional 5% levels. Could the authors also discuss the possibly of over-adjusting in the multivariable model, for example, adjusting for common confounders between the exposures & the outcomes?

* Could the authors provide more detail on the process of dealing with the extreme values. Was the truncation process applied to the creatinine measurements as well? Also, am I right that the measurements were not excluded, but rather it is their truncated values were used? How many measurements were truncated in the end?

Minor comments

* How was eGFR calculated from creatinine?

* Table 1: Inclusion criteria for RIISC: what does "with eGFR decline" mean?

[LINK]

---

## [Decision Letter · Decision Letter 1]

21 Jan 2020

Dear Dr. Fenton,

Thank you very much for re-submitting your manuscript "Association between non-malignant monoclonal gammopathy and adverse outcomes in chronic kidney disease: a cohort study" (PMEDICINE-D-19-03271R1) for review by PLOS Medicine.

I have discussed the paper with my colleagues and the academic editor and it was also seen again by xxx reviewers. I am pleased to say that provided the remaining editorial and production issues are dealt with we are planning to accept the paper for publication in the journal.

[LINK]

We look forward to receiving the revised manuscript by Jan 28 2020 11:59PM. 

Sincerely,

Adya Misra, PhD

Senior Editor 

PLOS Medicine

plosmedicine.org

Requests from Editors:

Author summary does not mention that the aim of the study was to compare the rates of MG in the general population with patients who have CKD. Since this is mentioned in the conclusions and discussion through the submission, please clarify if this information should be in the author summary

Please use p<0.001 throughout, ensuring p values are provided along with 95% CI

Some summary demographic information would be useful in the abstract 

Please remove spaces between refs where multiple ones cited 

Data availability- please clarify if the data underlying various graphs and tables can be provided as supplementary information as per PLOS Data policy. 

Introduction- Line 83 you say “MG is a prevalent condition”. It would be helpful to provide context as to its prevalence or a reference for our non specialist readers 

Comments from Reviewers:

Reviewer #1: I think the text has much improved including the new information and toning down conclusions.

Reviewer #2: The authors have addressed my concerns and I now recommend publication.

Peter Flom

Reviewer #3: The revised version of the manuscript 'Association between non-malignant monoclonal gammopathy and adverse outcomes in chronic kidney disease: a cohort study' by Dr Anthony Fenton and colleagues has improved very much. I still have some comments which the authors didi not respond.

The authors respondedto my comment that "Unfortunately, CRP data were not available. To our knowledge, there is no substantial body of evidence that demonstrates that infection or an elevated CRP is a significant predictor of MGUS progression." But, high CRP is a risk factor of the progression of CKD (Am J Kidney Dis. 2016 ;68(6):873-881, Ren Fail. 2015 ;37(1):45-9.). Since the authors investigated the progression of CKD by non-malignant monoclonal gammopathy, the role of CRP has to be evaluated. 

Reviewer #4: The authors have addressed all my comments, thank you. Study limitations have been acknowledged appropriately, together with the need for future research and suggestions and suggestions on which direction this research could take place in.

[LINK]

---

## [Editor Report · Decision Letter 2]

3 Feb 2020

Dear Dr Fenton, 

On behalf of my colleagues and the academic editor, Dr. Guiseppe Remuzzi, I am delighted to inform you that your manuscript entitled "Association between non-malignant monoclonal gammopathy and adverse outcomes in chronic kidney disease: a cohort study" (PMEDICINE-D-19-03271R2) has been accepted for publication in PLOS Medicine. 

PRODUCTION PROCESS

PRESS

PROFILE INFORMATION

Thank you again for submitting the manuscript to PLOS Medicine. We look forward to publishing it. 

Best wishes, 

Adya Misra, PhD

Senior Editor 

PLOS Medicine

plosmedicine.org